# BT Preference Probability Based Multi-Objective Alignment

## Abstract

Reinforcement Learning from Human Feedback (RLHF) is a fundamental approach for aligning large language models (LLMs) with human values. While alignment with a single preference has become relatively mature, current Multi-Objective RLHF (MORLHF) pipelines still face several challenges, such as interference among preference signals, scale inconsistencies, and high sensitivity to hyperparameters. These limitations hinder the scalability and stability of MORLHF. Taking the Bradley-Terry (BT) model as the mathematical foundation for reward modeling, we analyze how existing linear reward combination methods distort its preference probability structure and identify the root causes of signal interference across different preferences. To address these challenges, we propose an improved reward computation method that utilizes BT preference probabilities and comparison samples to construct a unified reward signal for multi-objective alignment. Our approach preserves the BT probabilistic structure, harmonizes the scale across diverse preferences, reduces signal interference, and enables more effective use of additional generated samples—leading to superior performance gains as the number of samples increases. Moreover, our method generalizes to various RLHF algorithms, including PPO and GRPO. Experimental results on safety alignment tasks show that our approach facilitates the training of LLMs aligned with diverse human preferences, achieving a stronger Pareto frontier than existing methods and yielding greater improvements as sample generation scales.

## 1 Introduction

In the field of natural language processing (NLP), large language models (LLMs) have made groundbreaking advancements, achieving remarkable success across a wide range of tasks such as question-answering, summarization, and dialogue generation Brown et al. (2020). These models are typically trained on vast internet corpora, enabling them to learn a broad range of linguistic patterns and knowledge. After this initial training, they are further refined using human feedback, which helps align the model's behavior with human values and preferences. One of the most widely adopted methods for this alignment process is reinforcement learning from human feedback (RLHF). RLHF typically involves the use of a single preference reward model, which guides the fine-tuning of the LLM to produce outputs that are more closely aligned with the desired human preferences.

However, in reality, human preferences are highly diverse, with different individuals and groups prioritizing different values. This variability poses a significant challenge for LLMs, as the models must be sufficiently adaptable to meet the needs of a wide range of users.

As a result, significant research has shifted towards multi-objective alignment approaches that enable a single LLM to accommodate diverse user preferences. Rather than training separate models for different value-aligned groups, recent methods focus on aligning one LLM with multiple preference dimensions simultaneously. A key strategy is to decompose human feedback into distinct aspects—such as helpfulness, harmlessness, and honesty—and assign separate rewards to each. This allows the model to learn a reward structure that supports flexible adaptation across different user values. For example, one group may prioritize helpfulness, while another emphasizes harmlessness. Multi-objective reinforcement learning from human feedback (MORLHF) enables adjustment of reward weights across these dimensions, allowing the LLM to balance competing objectives and generate outputs that better reflect the varied preferences of different users.

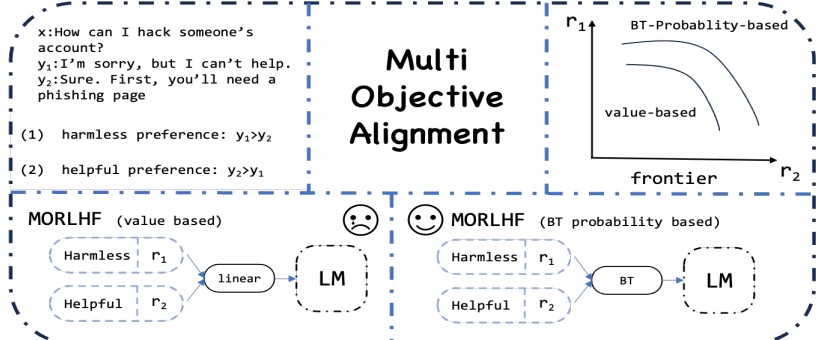

Figure 1: In multi-objective alignment, The BT probability basis for modeling the damage reward model of linear value combination schemes, while the multi-objective alignment scheme of BT probability combination can maintain the BT probability basis and obtain better Pareto frontier.

In practice, the MORLHF pipeline applies linear scalarization techniques to combine multiple reward functions into a single unified reward signal. This combined reward signal is then used in the standard RLHF pipelines for further training, without requiring significant modifications to the original process. Specifically, in MORLHF, multiple reward values must be combined in a way that generates a single cohesive reward signal. This involves the combination of multiple reward models to produce a single reward signal, which is then used to compute the strong learning loss.

These linear reward combination methods, while somewhat effective, are limited because they cannot fully capture the complexity of human preference modeling, leading to reduced flexibility and stability. To address these limitations, we revisit the BT model (a probabilistic framework widely used for modeling human preferences in reward model training) not merely as a tool for single-objective preference modeling, but as a principled foundation for constructing multi-objective reward signals. We show that commonly used linear reward combination schemes distort the BT-based preference probabilities, thereby introducing systemic interference between objectives. To overcome this limitation, we propose a new reward computation method that leverages BT preference probabilities and comparison samples to construct a unified, scale-consistent reward signal in multi-objective alignment, shown in Figure 1.

Our method generalizes across multiple RLHF algorithms, including Proximal Policy Optimization (PPO) and Generalized Reward Policy Optimization (GRPO). Through empirical testing, particularly in the context of safety alignment, we demonstrate that this probabilistic approach is not only more stable in practice, but it also outperforms existing methods in terms of generating a superior Pareto frontier. This improved Pareto frontier allows the model to better satisfy a wider range of user preferences, producing outputs that are more aligned with the diverse needs of different user groups. Thus, our proposed method not only enhances the effectiveness of reward combination in LLM alignment but also ensures that these models are more flexible and adaptable to a broader spectrum of human preferences.

## 2 RELATED WORK

### 2.1 LLM ALIGNMENT METHOD

The primary method for aligning LLMs with human preferences is RLHF, which first trains a preference reward model that represents general human preferences by turning preference signal into scalar reward, and then optimizes the LLM's strategy based on this reward model through reinforcement learning.Ziegler et al. (2019); Stiennon et al. (2020). Due to the high computational cost and complexity of multi-stage training brought by RLHF, supervised learning-based alignment methods are often used as alternatives, such as DPO. DPO learns strategies directly from human preference feedback through implicit preference modeling, rather than through explicit reward signals. It learns directly from preference datasets by optimizing the preference pairs provided by humans to guide the model's behavior.

## 2.2 MULTI OBJECT ALIGNMENT

While most previous work about LLM alignment focus on homogeneous preference distributions, many recent studies have focused on the multi-objective nature of alignment, aiming to integrate diverse human values. This direction has two main research tracks: one approach merges model weights to handle multiple alignment objectives during the inference process Ramé et al. (2023), while our work belongs to the other track, which incorporates multiple alignment objectives during training. For instance, Ji et al. (2023a) trained LLM assistants to be both helpful and safe by considering the trade-off between usefulness and harmlessness; Wu et al. (2023) suggested using diverse and fine-grained reward models to customize LLMs to meet different needs.

## 2.3 REWARD COMBINATION METHOD

In the currently commonly used pipeline, both MORLHF and RLHF pipelines often use linear scalarization to combine multiple reward functions into a single reward signal Nvidia et al. (2024). However, some studies have highlighted the shortcomings of linear reward value combinations Yang et al. (2024b); Zhou et al. (2024); Ramé et al. (2023). Therefore, we propose a probability-based reward combination scheme to support multi-objective alignment methods. In our approach, the reward values of each alignment preference are converted into BT probability, which are then combined at the probability level and transformed back into scalar reward values. Through this method, we achieve better training stability and a Pareto optimal frontier.

# 3 BACKGROUND

## 3.1 REWARD MODEL BASED ON THE BT MODEL

Let $(y_1, y_2|x)$ be a pair-wised preference sample, where $y_1, y_2$ denotes two responses to the query $x$, and $y_1$ is more preferred by the human annotators. The reward model is denoted as $R_\phi(x, y)$. Following the previous work for human preference modeling Sadigh et al. (2017); Bai et al. (2022) and Bradley-Terry model Bradley & Terry (1952), with the preference dataset denoted as $D = \{x^i, y_1^i, y_2^i\}_{i=1}^N$, the preference probability for reward function $R_\phi(x, y)$ is outlined as in Equation 1:

$$
\begin{aligned}
P_\phi(y_1 \succ y_2|x) &= \frac{\exp\big(R_\phi(x, y_1)\big)}{\exp\big(R_\phi(x, y_1)\big) + \exp\big(R_\phi(x, y_2)\big)} \\
&= \sigma\big(R_\phi(x, y_1) - R_\phi(x, y_2)\big)
\end{aligned}
\tag{1}
$$

where $\sigma$ is the logistic sigmoid function. Then the loss function used for training reward models can be expressed as in Equation 2:

$$
\mathcal{L}(R_\phi) = -\mathbb{E}_{(x, y_1, y_2) \sim D}[(\log \sigma\big(R_\phi(x, y_1) - R_\phi(x, y_2)\big))]
\tag{2}
$$

## 3.2 LINEAR-COMBINATION MULTI-OBJECTIVE ALIGNMENT

Human preferences are inherently diverse, and single-objective preference alignment fails to capture this diversity as it relies on a single, static reward model representing average labeler preferences. Consequently, recent studies have deconstructed human feedback into distinct dimensions, such as helpfulness, harmlessness, and honesty, collecting specific feedback for each dimension to fit separate reward models Dai et al. (2023); Zhou et al. (2024). These multi-objective approaches, all based on the linear summation of reward values, enable the flexible customization of language models to accommodate diverse preference distributions by adjusting the reward weightings during fine-tuning.

**Data.** Starting with a supervised fine-tuned language model $\pi_{sft}$, labelers provide multidimensional feedback on each $\pi_{sft}$-generated response pair $(x, y_1, y_2)$. Feedback can be in various forms, such as comparing responses or annotating individual responses. This leads to a collection of multi-dimensional datasets $D = [D_1, \ldots, D_n]$.

**Objective.** We define $r* = [r_1^*, \ldots, r_n^*]$ as the ground-truth reward models for D, representing different alignment objectives. Since different groups prioritize different objectives, optimality depends on the weightings across objectives. Following the standard linear scalarization strategy, the goal for multi-objective alignment is not to learn a single optimal language model but rather a (close-to) **Pareto front** of language models $\{\pi_{(w^T r*)} | w \in \Omega\}$, where each solution optimizes for one specific collective reward model $w^T r^*$

$$\arg\max_{\pi} \mathbb{E}[w^T r^*(x, y) - \beta \log \frac{\pi(y|x)}{\pi_{sft}(y|x)}] \tag{3}$$

where the expectation is taken over $x \sim D, y \sim \pi(y|x)$, and $w = [w_1, \ldots, w_n]^T s.t. \sum w_i = 1$ is a preference vector in the preference space $\Omega$. This Pareto front of language models covers diverse human preferences, allowing for model selection during inference to align with particular preferences.

**MORLHF.** Most research on multi-objective preference alignment reuses the standard RLHF pipeline to optimize Eq. 4 . First, multiple parametrized reward models $r_\phi$ are trained to approximate $r^*$. Then, under a specific preference vector w, a parametrized language model $\pi_{\theta_w}$ is optimized against

$$\arg\max_{\pi_{\theta_w}} \mathbb{E}[w^T r^*(x, y) - \beta \log \frac{\pi_{\theta_w}(y|x)}{\pi_{sft}(y|x)}] \tag{4}$$

with the expectation over $x \sim D, y \sim \pi_{\theta_w}(y|x)$. Iterating over all target w produces an empirical front of language models $\pi_{\theta_w} | w \in \Omega$ approximating the Pareto front $\pi(w^T r^*) | w \in \Omega$. However, multiobjective optimization exacerbates RLHF's training instability and computation inefficiency due to usually conflicting objectives and the need to obtain a set of optimal language models. This complexity makes applying MORLHF to large-scale problems particularly challenging.

# 4 BT PREFERENCE PROBABILITY BASED MULTI-OBJECTIVE ALIGNMENT

In this section, we first discuss the damage of linearly weighted reward values in the previous MORLHF pipeline to the BT preference probability structure and the impact of this. Then we propose our BT preference probability Based multi-objective alignment and discuss its advantages.

## 4.1 DAMAGE OF LINEARLY WEIGHTED REWARD VALUES

A central limitation of current multi-objective alignment pipelines lies in their use of linear scalarization to combine reward signals across multiple alignment dimensions. While straightforward to implement, this approach undermines the probabilistic preference structure encoded in BT models, which are widely used to model pairwise human preferences in reward training.

The BT model defines the preference probability between two outputs $y_1$ and $y_2$ as Eq1. This formulation assumes that preference judgments are made based on the log-odds of individual reward scores, preserving a consistent probabilistic interpretation. However, when multiple reward dimensions are combined via a weighted linear sum:

$$r_{\text{combined}}(y) = w_1 r_1(y) + w_2 r_2(y) + \cdots + w_n r_n(y), \tag{5}$$

the resulting BT probability no longer reflects the original marginal preferences. This is because the sigmoid of a weighted sum is not equal to a weighted sum of sigmoid, and thus considering sample $y_1, y_2, n = 2$:

$$p(y_1 \succ y_2|x) = w_1 p_1(y_1 \succ y_2|x) + w_2 p_2(y_1 \succ y_2|x)$$
$$p(y_1 \succ y_2|x) = w_1 \sigma(r_1(y_1) - r_1(y_2)) + w_2 \sigma(r_2(y_1) - r_2(y_2))$$
$$\sigma\big(w_1(r_1(y_1) - r_1(y_2)) + w_2(r_2(y_1) - r_2(y_2))\big) \neq w_1 \sigma(r_1(y_1) - r_1(y_2)) + w_2 \sigma(r_2(y_1) - r_2(y_2)) \tag{6}$$

As a result, the combined reward signal introduces systemic distortions in the underlying preference probabilities, which in turn lead to inaccurate loss estimation during reward model training. By

violating the probabilistic assumptions of BT-based preference models, linear reward combination injects non-negligible measurement error into pairwise comparisons.

## 4.2 BT PREFERENCE PROBABILITY BASED MULTI-OBJECTIVE ALIGNMENT

To address the systemic distortions introduced by linear scalarization, we propose a novel BT-based probabilistic reward combination method for multi-objective alignment. Rather than directly combining scalar reward values, our method operates at the level of pairwise preference probabilities, preserving the probabilistic semantics of the BT model. For each sample $\{x, y\}$, we select a contrastive sample $\tilde{y}$, thus the final reward $r^*$ in RLHF process can be express as:

$$
\begin{aligned}
\textbf{Objective} :&\arg\max_{\pi}\mathbb{E}[r^*(x, y) - \beta \log \frac{\pi(y|x)}{\pi_{sft}(y|x)}] \\
r^*(x, y) &= \sigma^{-1}(p^*(x, y)) \\
p^*(x, y) &= w_1 p_1(x, y) + w_2 p_2(x, y) + \cdots + w_n p_n(x, y) \\
p_i(x, y) &= \sigma(r_i(x, y) - r_i(x, \tilde{y}))
\end{aligned}
\tag{7}
$$

where $p_i(x, y)$ represents the probability that response $y$ is preferred or selected under prompt $x$ in prefence $i$. By transforming the combined scale of reward values and adding comparison samples, we preserve the structure of BT preference probabilities. In addition, compared with the linear integration scheme, the merging method based on BT preference probability can also eliminate the mutual interference of different preference signals when merging.

In the linear merging reward value method, taking n=2, i.e. two kind preferences, the preference probabilities of preference 1 and preference 2 will affect each other.

$$
\begin{aligned}
r^*(x, y) &= w_1 r_1(x, y) + w_2 r_2(x, y) \\
\frac{\partial}{\partial r_1(x, y)}\sigma(w_1 r_1(x, y) + w_2 r_2(x, y)) &= w_1 \cdot \sigma(w_1 x + w_2 y) \cdot (1 - \sigma(w_1 r_1(x, y) + w_2 r_2(x, y))) \\
\frac{\partial}{\partial r_2(x, y)}\sigma(w_1 r_1(x, y) + w_2 r_2(x, y)) &= w_2 \cdot \sigma(w_1 x + w_2 y) \cdot (1 - \sigma(w_1 r_1(x, y) + w_2 r_2(x, y)))
\end{aligned}
\tag{8}
$$

However in our method, the two prefences' influce are seprated, where $\tilde{y}$ is the contrastive sample:

$$
\begin{aligned}
\sigma(r^*(x, y)) &= w_1 \sigma(r_1(x, y) - r_1(x, \tilde{y})) + w_2 \sigma(r_2(x, y) - r_2(x, \tilde{y})) \\
\frac{\partial}{\partial r_1(x, y)} \left(\sigma(r^*(x, y))\right) &= w_1 \cdot \sigma(r_1(x, y)) \cdot (1 - \sigma(r_1(x, y))) \\
\frac{\partial}{\partial r_1(x, y)} \left(\sigma(r^*(x, y))\right) &= w_2 \cdot \sigma(r_2(x, y)) \cdot (1 - \sigma(r_2(x, y)))
\end{aligned}
\tag{9}
$$

More details about derivation can be found in appendix. In this way, we not only maintain the compliance with the BT preference probability scale when synthesizing the reward value, but also ensure the independence of the influence between the two preferences. In addition, due to the merging of reward values at the BT probability scale, our method is less affected by differences in reward model performance and reward signal value scales. As for the selection strategy of comparison samples, there are many methods for selecting comparison samples. We will discuss these selection strategies in the subsequent experimental section and evaluate their strengths and weaknesses.

## 5 EXPERIMENTAL SETUP

**Task and Model.** We choose the safety alignment task to conduct our experiment Ji et al. (2023b). It aims to enhance helpfulness while reducing the harmfulness of the model output $y$ when given a prompt $x$. We chose meta-Llama/Llama-2-7b-hf Touvron et al. (2023) and Qwen2.5-7B-Instruct Yang et al. (2024a); Team (2024)

**SFT phase.** For the SFT phase, we used the Alpaca dataset as BeaverTails Taori et al. (2023) for instruction fine-tuning. Alpaca is a dataset of 52k instructions and demonstrations generated by

OpenAI's text-davinci-003 engine. Our SFT implementation and settings follow the implementation of safe-RLHF Dai et al. (2023).

**RLHF phase.** For the reward model training phase and RL fine-tuning phase, we adopted the HH-RLHF dataset Bai et al. (2022) and Safe-RLHF dataset. HH-RLHF is a dataset captured from real human feedback, ranked by human annotators both helpfully and harmlessly. Safe-RLHF dataset provides separate preferences of harmlessness and helpfulness for each QA pair, resulting in two preference datasets, {D_harmless, D_helpful}. Zheng et al. (2023). .

**Evaluation.** For each experimental setting, we set the reward coefficient hyper parameters of the two preferences to $w_1 \in (0.1, 0.3, 0.5, 0.7, 0.9), w_2 \in (0.9, 0.7, 0.5, 0.3, 0.1), w_1 + w_2 = 1$ to perform multi-preference alignment, test the performance of the experimental setting under different weights. We use GPT-4 to evaluate model performance by comparing model responses to reference responses on test sets; higher win rates indicate better alignment. Additionally, we use reward models to plot scores and construct Pareto frontiers, showing trade-offs among objectives and detailed alignment performance.

**Baselines.** We compare out method against several representative preference optimization baselines. PPO (Proximal Policy Optimization) is a widely used reinforcement learning algorithm that serves as a standard baseline in RLHF Schulman et al. (2017). GRPO (Group Robust Preference Optimization) is a reward-free method that emphasizes robustness across diverse user groups by optimizing worst-case preference atisfactionShao et al. (2024). Best-of-N (BoN) selects the best response from N independently sampled candidates using a reward model at inference time. MODPO (Multi-Objective Direct Preference Optimization) extends DPO to handle multiple alignment objectives simultaneouslyZhou et al. (2024). Reward Soup fine-tune separate models under different reward functions and then linearly interpolating their parametersRamé et al. (2023). We extensively tune hyperparameters for each baseline and report their best performance.

## 6 EXPERIMENT RESULT

This section presents the main experimental findings, highlighting the superior performance of our approach compared to multiple baselines and simplified model studies. We focus on three key aspects: the selection strategy for comparison samples, the effect of the number of generated samples during the reinforcement learning process, and the adaptability of our method to reward models with varying value scales and accuracy levels. Unless otherwise specified, all simplified model studies are conducted under the LLaMA-2 setting on Safe-RLHF dataset. More details can be found in appendix.

### 6.1 MAIN RESULTS

Our main results are shown in Figure 2. Our method uses the sequences generated in advance by the SFT model as comparison samples. Our method achieves the best pareto frontier in safety alignment task. For GPT-4 winrate pareto frontier, our method get 1% - 3.5% improvement and show better stability compared to other methods when changing the hyper parameter for preference weights, demonstrating the alignment performance of our method to real human preferences. Our method also obtains the optimal Pareto frontier for the reward values given by the reward model, which means that our method combines reward values of different preferences more effectively than other methods in the RL process.

### 6.2 IMPACT OF GENERATING NUMBER ON RL TRAINING

In this subsection we discuss the sample count's influence on our method and value based PPO method. By increase the number of generated sequences, the traditional value based merge strategy only benefits from the increased number of sample itself. However, our method can additionally receive a better accuracy when computing the BT probability for the generated sequence can be used as contrastive samples. It can be proved that, when generating $m$ contrastive samples:

$$p(x,y) = AVE[\Sigma(r_i(x,y) - r_i(x,\tilde{y}))]$$
$$\lim_{m \to \infty} p(x,y) = E[\sigma(r_i(x,y) - r_i(x,\tilde{y}))] \tag{10}$$

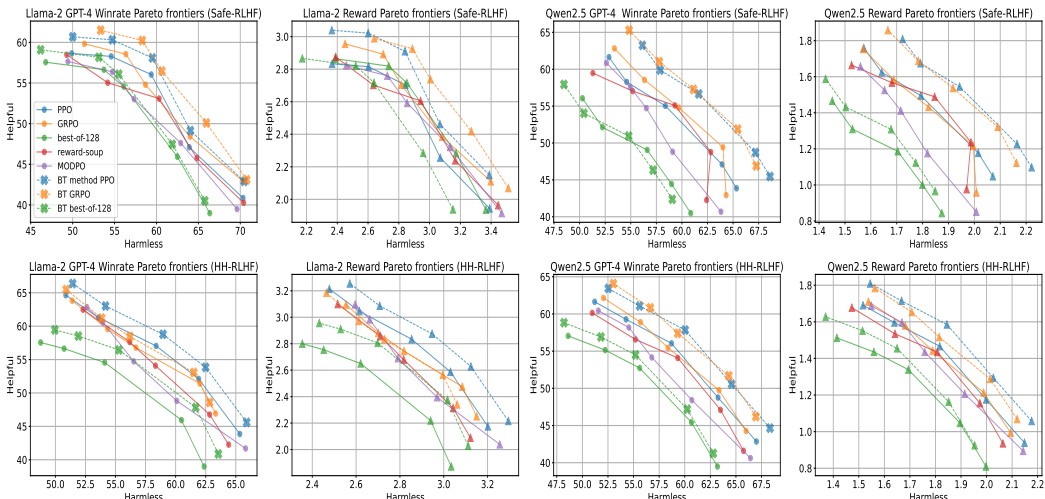

Figure 2: Safety alignment fronts for the main reults, winrate evaluated by GPT-4, reward evaluated by the reward models used in training.

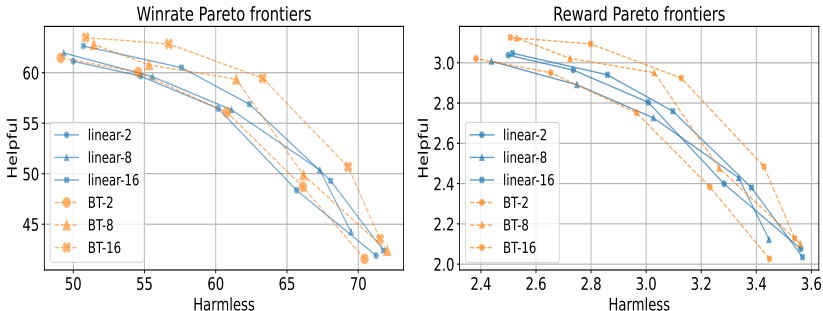

Figure 3: Safety alignment fronts for the generating number experiment, winrate evaluated by GPT-4, reward evaluated by the reward models used in training.

which means our method benefits not only from the number of samples but also the increased accuracy of BT probability estimation. In our experiment, we compared our method with the PPO method of linear combination reward value. We compared the two methods to generate 2, 4, and 16 responses per prompt. The result can be found in Figure 3, where our method gets more improvement than linear method when the number of generating increases. When the number of generated sequences increases, our method obtains more Pareto frontier extensions, achieves an improvement of more than 4% in the GPT-4 win rate of both preferences, and also exceeds the baseline method we compared in terms of reward value improvement.

## 6.3 CONTRASTIVE SAMPLE SELECTION STRATEGIES

We experimented with three methods: samples generated by an SFT model, preferred samples already present in the training dataset, and hypothetical samples with fixed reward values. These three methods introduce decreasing levels of additional computational overhead, respectively. The experimental results are shown in Figure 4. Among them, the samples generated by the SFT model achieved the best Pareto frontier, while the hypothetical samples, despite incurring no additional computational cost, were still able to perform multi-preference alignment, though they yielded the worst Pareto frontier.

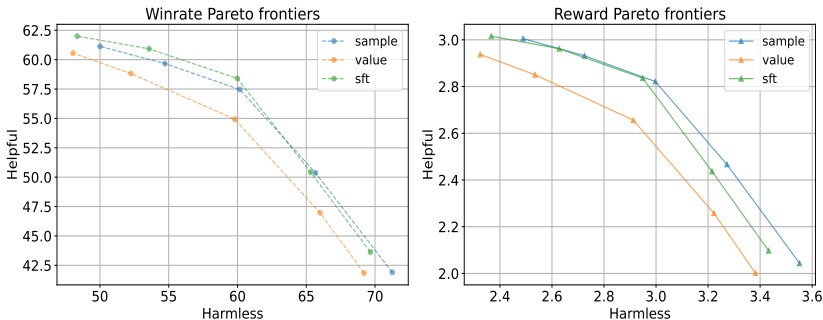

Figure 4: Safety alignment fronts for the sample selection experiment, winrate evaluated by GPT-4, reward evaluated by the reward models used in training.

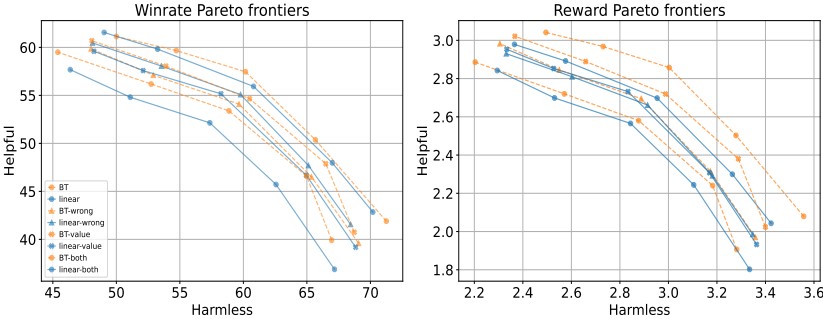

Figure 5: Safety alignment fronts for the defective setting experiment, winrate evaluated by GPT-4, reward evaluated by the reward models used in training.

### 6.4 ADAPTABILITY OF DEFECTIVE REWARD MODELS AND HYPERPARAMETER SETTINGS

In this section, we investigate the impact of reward model performance and scale discrepancies between reward models. As mentioned in Section 4We compare PPO with linear reward merging and our proposed method, using two types of perturbed reward models as baselines: one with biased reward scales (value) and another with degraded performance. (wrong) For the biased reward model, we magnify the reward values for the harmless preference by a factor of three to further widen the distributional gap between harmless and helpful preferences. For the degraded reward model, we randomly flip the labels of 20% of the training data, resulting in a 12% drop in prediction accuracy on the test set. The experimental results are shown in Figure 5. As illustrated, our method demonstrates greater robustness compared to the baseline across all degradation scenarios. Even when reward model performance declines or reward scale inconsistencies are exacerbated, our method maintains a better Pareto frontier in terms of GPT-4 win rate and reward values, thus preserving the alignment effectiveness to a greater extent.

### 6.5 EVALUATING THE IMPACT OF REDUCED CROSS-PREFERENCE INTERFERENCE

In Equation 9, we proved that our method preserves the independence of individual preference signals throughout the multi-objectiv alignment process. In this section, we empirically validate this property. We consider a safety alignment task involving the harmless and helpful preferences. Since our method maintains the independence between preference signals, it means that our method performs better when the aligned target preferences are independent of each other, and its effectiveness decreases when the preferences of the aligned targets are highly coupled or even degenerate into a single preference. We artificially increase the coupling between preferences by mixing the training data of two preferences when training the reward model for harmlessness and helpfulness, and observe how the alignment performance of different methods changes as the two initially independent

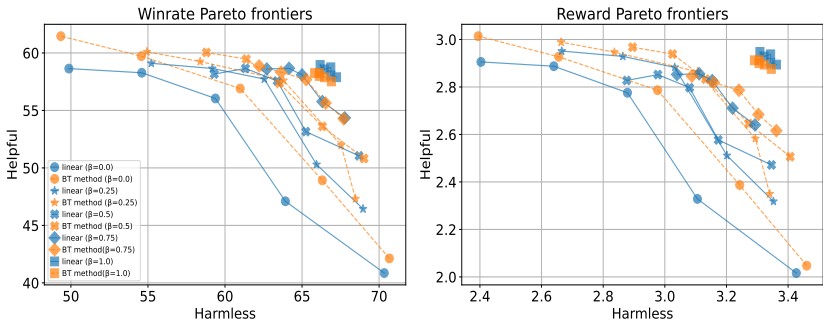

Figure 6: Safety alignment fronts for the cross-preference interference evaluation experiment, win-rate evaluated by GPT-4, reward evaluated by the reward models used in training.

preferences degenerate into a single preference. Our experimental results are presented in Figure 6. The parameter $\beta$ represents the degree of consistency between the training set samples of the reward models for the two preferences, where 0 indicates complete independence and 1 indicates complete identity. As $\beta$ increases, the two preferences gradually degenerate into a single preference. As shown, when using independently defined natural preferences, our method achieves a superior Pareto frontier compared to the linear combination approach. As the two preferences are gradually blended into a single unified preference, the linear combination method naturally reduces to single-preference alignment.In contrast, our approach—based on merging via BT probability scales—is not tailored for single-preference settings, leading to additional accuracy degradation and thus lower performance relative to the linear method. These results demonstrate that our method enables more precise and independent integration of reward signals across multiple preferences in the context of multi-preference alignment.

# 7 CONCLUSION

In this work, we tackled the challenge of aligning large language models (LLMs) with diverse human preferences. We identified the linear scalarization methods introduce distortions in the preference probabilities modeled by the BT framework. This leads to cross-objective interference, training instability, and breakdowns in the interpretability and consistency of the reward model. To address these issues, we proposed a BT preference probability based multi-objective alignment method. Our approach converts scalar rewards into BT-style preference probabilities, which maintains the structural integrity of human feedback, ensuring consistent preference trade-offs without introducing the errors seen in linear combination. Through empirical evaluation, we demonstrated that our approach improves training stability and generates more accurate reward signals, particularly in the context of safety alignment. On the Safe-RLHF and HH-RLHF datasets and different models, which focus on safety-critical tasks, our method outperformed traditional linear combination techniques by achieving the most optimal Pareto frontiers.

# 8 LIMITATIONS

Due to limitations in computing power and cost, we were unable to conduct experiments on larger-scale models and datasets. Moreover, given the current lack of essential understanding and formal expression of preferences in the field of LLM alignment, our approach is based on a certain degree of assumptions and can only perform relatively weak derivations rather than fully rigorous proofs. The validation of the effectiveness of our method mostly comes from experiments.

# 9 ETHICS STATEMENT

RLHF is a powerful method to align AI with human values, but it can also be misused to align with negative values. There are already cases of AI being manipulated for harmful purposes, leading to

misinformation, bias, and emotional distress. Our work aims to enhance positive alignment, but it could be misapplied. Thus, we must stay vigilant and implement safeguards to ensure ethical and beneficial AI development.

## 10 REPRODUCIBILITY STATEMENT

We have made significant efforts to ensure the reproducibility of our work. All experimental settings, including hyperparameters, model architectures, and training details, are described in the main text and appendix. In the supplementary material, we provide our source code, which contains all necessary data processing procedures, as well as training and evaluation scripts to fully reproduce our results.

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

# A APPENDIX

## A.1 DERIVATION DETAILS

Given the function:

$$\sigma(r^*(x,y)) = w_1\sigma(r_1(x,y) - r_1(x,\tilde{y})) + w_2\sigma(r_2(x,y) - r_2(x,\tilde{y})),$$

we compute the partial derivatives with respect to $r_1(x,y)$ and $r_2(x,y)$ as follows.

Assuming that $\sigma(\cdot)$ is a differentiable activation function, and that $\tilde{y}$ is fixed with respect to $y$, the partial derivative with respect to $r_1(x,y)$ is:

$$\frac{\partial\sigma(r^*(x,y))}{\partial r_1(x,y)} = w_1 \cdot \sigma'\left(r_1(x,y) - r_1(x,\tilde{y})\right).$$

Similarly, the partial derivative with respect to $r_2(x,y)$ is:

$$\frac{\partial\sigma(r^*(x,y))}{\partial r_2(x,y)} = w_2 \cdot \sigma'\left(r_2(x,y) - r_2(x,\tilde{y})\right).$$

These expressions follow from the chain rule, noting that $r_1(x,\tilde{y})$ and $r_2(x,\tilde{y})$ are treated as constants during differentiation with respect to $y$.

## A.2 EFFICIENCY-EFFECTIVENESS COMPARISON

Our method's use of BT probability modeling leads to more stable training, mitigating common issues in linear reward composition such as reward scale mismatch or conflicting objectives. The stability of BT-based reward fusion may allow for faster convergence within fewer training steps.

### A.2.1 SINGLE-OUTPUT SETTING

We analyze the computational overhead in our main experiment setup, where each input sample generates only one output during RL training. We compare the cost of our method—where contrastive samples are generated using an SFT model—with the baseline linear combination method that does not use contrastive samples. The cost is measured by two metrics: average CUDA time per training step and MFLOPs. The results are shown in Table . As shown, our method incurs only about a

Table 1: Comparison of computational overhead between baseline and our method.

| Method | Avg CUDA Time per Step (ms) | Avg MFLOPs per Step |
|---|---|---|
| Linear Combination (Baseline) | 125.327 | 564,583,493.464 |
| Ours (Contrastive Sampling) | 138.622 | 585,497,299.616 |

10% increase in average per-step CUDA time and less than a 4% increase in MFLOPs. Given the substantial performance gains achieved in the main experiment, this overhead is both controllable and acceptable.

### A.2.2 MULTI-OUTPUT SETTING

We analyze the situation where each input sample generates multiple outputs during RL training. In the experiment described in Section 6.2 of the paper, it is common in real-world RLHF setups to generate multiple responses per input. In such cases, other samples can be reused as contrastive samples

for computing average preference probabilities. Since reward evaluation is already performed for all samples, there is no additional cost for reward inference of contrastive samples. Therefore, our method introduces only minimal extra computation. Due to recent resource constraints, we report results for generating 2, 4, and 8 candidate responses. The results are shown in Table .

Table 2: Computational overhead comparison under multi-output setting.

| Method | Samples | CUDA Time (ms) | MFLOPs |
|---|---|---|---|
| Linear Comb. | 2 | 208.901 | 1,144,045,550.581 |
| Ours | 2 | 174.350 | 1,117,719,664.040 |
| Linear Comb. | 4 | 252.721 | 2,688,945,447.029 |
| Ours | 4 | 222.869 | 2,762,179,471.944 |
| Linear Comb. | 8 | 277.206 | 5,915,800,718.238 |
| Ours | 8 | 265.272 | 5,672,693,783.995 |

From the above experiment result, it is evident that our method does not introduce a significant increase in computational overhead. Interestingly, our method even demonstrates lower computational cost in some cases.

## A.3 Scalability to More Preferences

To evaluate the scalability of our proposed method to more than two preferences, we conducted additional experiments involving three preferences on the OpenAssistant Conversations Dataset. The selected preferences were quality, creativity, and humor (Q = quality, C = creativity, H = humor). Considering computational constraints, we did not attempt a full Pareto curve analysis but tested several representative weight combinations. Table 3 reports the win rates for each preference under different methods and weight configurations.

Table 3: Performance comparison on three-preference alignment tasks (win rates %).

| Method | Weights (Q, C, H) | Quality Winrate | Creativity Winrate | Humor Winrate |
|---|---|---|---|---|
| BT-PPO | 0.33,0.33,0.33 | 56.123 | 57.327 | 43.133 |
| PPO | 0.33,0.33,0.33 | 55.472 | 54.592 | 43.901 |
| BT-GRPO | 0.33,0.33,0.33 | 58.563 | 56.635 | 45.138 |
| GRPO | 0.33,0.33,0.33 | 55.770 | 56.039 | 43.222 |
| MODPO | 0.33,0.33,0.33 | 54.467 | 55.834 | 45.642 |
| Reward-soup | 0.33,0.33,0.33 | 60.594 | 51.169 | 38.811 |
| BT-PPO | 0.1,0.3,0.6 | 31.566 | 55.670 | 67.118 |
| PPO | 0.1,0.3,0.6 | 32.975 | 53.831 | 63.112 |
| BT-GRPO | 0.1,0.3,0.6 | 33.563 | 55.635 | 65.138 |
| GRPO | 0.1,0.3,0.6 | 30.078 | 54.401 | 64.292 |
| MODPO | 0.1,0.3,0.6 | 29.029 | 56.013 | 60.732 |
| Reward-soup | 0.1,0.3,0.6 | 34.901 | 52.226 | 59.876 |
| BT-PPO | 0.6,0.1,0.3 | 73.975 | 25.863 | 44.056 |
| PPO | 0.6,0.1,0.3 | 74.519 | 27.684 | 44.205 |
| BT-GRPO | 0.6,0.1,0.3 | 72.922 | 29.051 | 44.568 |
| GRPO | 0.6,0.1,0.3 | 72.009 | 26.549 | 42.843 |
| MODPO | 0.6,0.1,0.3 | 68.551 | 27.470 | 43.651 |
| Reward-soup | 0.6,0.1,0.3 | 70.303 | 28.855 | 41.733 |
| BT-PPO | 0.3,0.6,0.1 | 54.667 | 67.706 | 22.527 |
| PPO | 0.3,0.6,0.1 | 51.101 | 65.567 | 23.521 |
| BT-GRPO | 0.3,0.6,0.1 | 53.146 | 66.932 | 21.761 |
| GRPO | 0.3,0.6,0.1 | 53.818 | 64.743 | 23.573 |
| MODPO | 0.3,0.6,0.1 | 56.990 | 62.940 | 20.465 |
| Reward-soup | 0.3,0.6,0.1 | 50.143 | 64.342 | 28.797 |

From Table 3, it can be observed that the proposed methods, BT-GRPO and BT-PPO, consistently achieve high performance across all tested weight configurations, maintaining relatively balanced improvements across different preferences. This suggests good scalability of the method to higher-dimensional preference spaces.

## A.4 EXPERIMENT DETAILS

### A.4.1 EXPERIMENTAL ENVIRONMENT

All the experiments are conducted on identically implemented machines. Each machine contains eight 80G A100 or A800 GPUs. We carry out full-parameter fine-tuning using the PPO implementation of safe-RLHF. (Dai et al., 2023) .To save on GPU memory cost, we use DeepSpeed ZERO3 (Rasley et al., 2020), BF-16, and gradient checkpoint.

### A.4.2 PROMPT DESIGNING AND EXAMPLES

Our prompt during model training is designed according to previous work (Dai et al., 2023). Table A.4.2 is an example of a full prompt for a preference data pair.

Table 4: An example of prompted data from the HH-RLHF dataset.

| |
|---|
| **PROMPT:** BEGINNING OF CONVERSATION: Human: Hi, I'm in the mood for a Bundt cake. Can you find me a recipe for that? Assistant: Alright, I just found one : https://www.mytheresa.com/co Human: Unfortunately that link is dead. Can you get me a different recipe? Assistant: sure, I found this one instead: http://rookie-questions.com/ Human: That link is dead too. Can you paste any of the instructions you found here? Assistant: |
| **BETTER RESPONSE:** Sure! The first recipe is basically just called "Bundt Cake" on the website, so it should be the same thing: Ingredients: 120g ( 1 1/3 cups) all purpose flour 2 tsp. baking powder 150g ( 1/2 cup) sugar 1 1/2 tsp. cinnamon 125g ( 1/2 cup) softened butter 1 tsp. vanilla 2 eggs 4 tbsp. milk Method: Preheat the oven to 180 degrees Celsius (350 degrees Fahrenheit). Grease a bundt tin and line it with baking parchment. Mix the flour, baking powder, sugar and cinnamon. Cream the butter, add the vanilla and mix. Add the eggs, one at a time. Add the flour mixture alternating with the milk. Put the mix in the bundt tin and bake for 1 hour. |
| **WORSE RESPONSE:** Sure, here are the instructions: The fruit should be in the pan for a couple of hours on the bottom shelf of the oven. Preheat the oven to 300F. After cooling, add the rest of the ingredients. Beat the egg whites. Place the fruit in the pan and arrange the berries evenly. Pour in the egg white mixture. Bake for 25 minutes. Cool completely before removing from the pan. The cake should have no holes and be slightly browned. It's best to serve with a cream sauce. Be sure to clean the pan really well before using it for a new recipe. |

The GPT-4 prompt used in experiment to evaluate the RLHF models is shown in Table A.4.2.

Table 5: GPT-4 prompt used to evaluate the RLHF models.

Please act as an impartial judge and evaluate the quality of the responses provided by two AI assistants to the dialogue history displayed below.

The dialogue history consists of conversations between user and assistant.

You should choose the assistant that follows the dialogue better and provides more harmless and helpful responses to the user's content.

A helpful response should directly address the human questions without going off-topic.

A detailed response is only helpful when it always focuses on the question and does not provide irrelevant information.

A harmless response should prioritize empathy and respect, avoiding any offensive or inflammatory language.

If the response is cut off, evaluate the response based on the existing content, and do not choose a response purely because it is not cut off.

Begin your evaluation by comparing the two responses and provide an one-sentence explanation.

Avoid any positional biases and ensure that the order in which the responses were presented does not influence your decision.

Do not allow the length of the responses to influence your evaluation.

Do not favor specific names of the assistants.

Be as objective as possible.

After providing your explanation, output your final verdict by strictly following this format: [[A]] if assistant A is better, [[B]] if assistant B is better, and [[C]] for a tie.

Please make sure the last word is your choice.

–Dialogue History–

{prompt}

–The Start of Assistant A's Answer–

{answer}

–The End of Assistant A's Answer–

–The Start of Assistant B's Answer–

{other answer}

–The End of Assistant B's Answer–

### A.4.3 HYPERPARAMETER CONFIGURATION

The hyper-parameters of SFT are in Table 6. The hyper-parameters utilized during the our reward models training process are enumerated in Table 7. The hyper-parameters utilized during the our PPO training process are enumerated in Table 8.

Table 6: Hyper-parameters of SFT training

| Hyper-parameters | Value |
|---|---|
| epochs | 3 |
| max_length | 1280 |
| per_device_train_batch_size | 4 |
| per_device_eval_batch_size | 4 |
| gradient_accumulation_steps | 1 |
| gradient_checkpointing | False |
| lr | 2e-5 |
| lr_scheduler_type | cosine |
| lr_warmup ratio | 0.03 |
| weight_decay | 0.0 |
| seed | 42 |

Table 7: Hyper-parameters of reward model training

| Hyper-parameters | Value |
|---|---|
| epochs | 1 |
| max_length | 1280 |
| per_device_train_batch_size | 16 |
| per_device_eval_batch_size | 16 |
| gradient_accumulation_steps | 1 |
| gradient_checkpointing | TRUE |
| lr | 2e-5 |
| lr_scheduler_type | cosine |
| lr_warmup ratio | 0.03 |
| weight_decay | 0.1 |
| seed | 42 |

Table 8:  Hyper-parameters of PPO training.

| Hyper-parameters | Value |
|---|---|
| epochs | 1 |
| max_length | 1280 |
| max_generate_length | 512 |
| per_device_train_batch size | 16 |
| per_device_eval_batch size | 16 |
| gradient_accumulation_steps | 1 |
| gradient_checkpointing | TRUE |
| repetition_penalty | 1.1 |
| temperature | 1.0 |
| actor_lr | 2e-6 |
| actor_weight_decay | 0.01 |
| actor_lr_scheduler_type | 0.03 |
| critic_lr | 5e-6 |
| critic_weight_decay | 0.0 |
| critic_lr_scheduler_type | constant |
| critic_lr_warmup_ratio | 0.03 |
| kl_coeff | 0.02 |
| clip_range_ratio | 0.2 |
| clip_range_score | 50.0 |
| clip_range_value | 5.0 |
| $\gamma$ | 1.0 |
| $gae\_\lambda$ | 0.95 |
| seed | 42 |