# OpenReview forum: "Probability-based Reward Value Combination Method for Multi-Objective Alignment"
_ICLR.cc/2026/Conference — Submitted to ICLR 2026_

### Official Review · Reviewer_zcFE · 2025-10-28

**Soundness:** 2
**Presentation:** 2
**Contribution:** 2
**Rating:** 2
**Confidence:** 4

**Summary:**

This paper proposes a new approach to multi-objective alignment in reinforcement learning from human feedback (RLHF), called Probability-Based Reward Composition (PBRC). The authors argue that existing methods, which combine multiple reward models using fixed linear weights, break the probabilistic structure assumed by the Bradley–Terry (BT) preference model commonly used in reward learning.

PBRC instead performs the combination at the probability level rather than at the raw reward level. Each reward model first produces a preference probability between responses, and these probabilities are then combined through weighted averaging before being converted back into a single composite reward. This formulation is intended to maintain internal probabilistic consistency across objectives such as helpfulness, harmlessness, and honesty.

The proposed method is integrated into standard RLHF or GRPO pipelines for policy optimization. Experiments on alignment tasks show that PBRC provides more stable training, improved trade-offs between competing objectives, and better Pareto efficiency compared with conventional linear scalarization. The paper positions PBRC as a theoretically grounded framework for consistent and interpretable multi-objective reward modeling.

**Strengths:**

The paper is original in its motivation and framing. It identifies an important limitation of current multi-objective RLHF methods that linearly combining multiple reward signals breaks the probabilistic structure implied by the Bradley–Terry (BT) preference model. The proposed Probability-Based Reward Composition (PBRC) shifts the combination from reward values to preference probabilities, which is a creative and theoretically motivated idea. This reframing represents a novel attempt to improve the internal consistency of multi-objective alignment.

In terms of technical quality, the idea is theoretically interesting but not fully developed. For significance, the problem addressing probabilistic consistency in multi-objective alignment is important, and the proposed direction has potential. However, the current presentation and incomplete theoretical treatment prevent the paper from realizing its full impact.

**Weaknesses:**

1. Major Weakness – Inconsistency with Target BT Formulation:
The paper aims for a composite reward $r^\ast$ such that the aggregated preference probability satisfies
$$p(y_1 \succ y_2 \mid x) = \sigma\(r^\ast(x, y_1) - r^\ast(x, y_2))$$
ensuring Bradley--Terry (BT) consistency at the multi-objective level.
However, in Equation (7), the way $r^\ast$ is constructed is inconsistent with this goal.


2. The clarity and overall presentation of the paper are poor. Many equations are introduced without sufficient explanation, and the notation is inconsistent throughout.  For example, line 176 refers to Eq. 4, while line 199 refers to Eq1, suggesting inconsistent formatting.
Additionally, lines 291--292 lack spaces before cited papers. These inconsistencies make the paper difficult to follow. A careful revision of notation, equation references, and citation formatting is strongly recommended to improve readability and professionalism.

**Questions:**

1. Clarification on Equation (7): Could the authors provide a detailed derivation and explanation of Equation (7)?
Specifically, how does this formulation resolve the inconsistency identified in Equation (6)?
It is unclear how the proposed construction ensures BT consistency across multiple objectives.
From the current description, it appears that
$$p(y_1 \succ y_2 \mid x) \neq \sum_i w_i \, p_i(y_1 \succ y_2 \mid x)$$
still holds. A more rigorous explanation or proof of how Equation (7) restores BT consistency would significantly strengthen the paper’s theoretical contribution.

2. Role of $\tilde{y}$ in Equation (7): Could the authors clarify the role of the contrastive response $\tilde{y}$ in Equation (7)?
It is not explicitly stated what $\tilde{y}$ represents or whether it corresponds to the dispreferred sample $y^-$ in the preference pair.
Understanding the precise meaning and function of $\tilde{y}$ is essential for interpreting how the composite reward $r^\ast(x, y)$ is computed and how it interacts with the BT formulation.

3. Equations (8) and (9) introduce an assumption of independence between reward models, could the authors explain why this independence is required? During model optimization, the gradients should normally be computed with respect to the parameters of the policy rather than the reward outputs. It is unclear what practical meaning or theoretical justification the derivative with respect to the rewards provides in this context.  Please explain the purpose of this differentiation and how it fits into the optimization process of the proposed method. Moreover second line of Equation 9 has a typo.

---

### Official Review · Reviewer_xYKq · 2025-10-30

**Soundness:** 3
**Presentation:** 3
**Contribution:** 2
**Rating:** 4
**Confidence:** 3

**Summary:**

This paper introduces a BT-preference-probability-based multi-objective alignment method for LLMs, replacing traditional linear reward combinations with probabilistic fusion of human preference signals to preserve the Bradley–Terry structure—achieving greater stability, and a stronger Pareto frontier across multiple objectives in RLHF.

**Strengths:**

1. The paper presents an original probabilistic formulation for multi-objective alignment that replaces the conventional linear scalarization with a BT-based preference probability model. This idea is both conceptually novel and methodologically rigorous, preserving the probabilistic semantics of human feedback while addressing long-standing issues of signal interference and scale mismatch in RLHF training.

2. The paper is clearly written, with strong theoretical motivation, detailed mathematical derivation, and comprehensive empirical validation across multiple models and datasets. Its significance lies in providing a principled and generalizable framework that improves the stability and scalability of aligning LLMs with diverse human preferences.

**Weaknesses:**

1. The experiments are restricted to medium-scale models (LLaMA-2-7B and Qwen2.5-7B) and safety-alignment datasets. Although the authors acknowledge computational limitations, the lack of evaluations on larger-scale models and higher-quality or more diverse datasets limits the assessment of the method’s scalability and its potential for broader application.

2. From the overall contribution perspective, the paper remains somewhat limited in scope. It does not explore preference fusion beyond human feedback signals—for instance, integrating Reinforcement Learning with Verifiable Rewards (RLVR).

**Questions:**

1. How was the reported improvement of 1%–3.5% in the GPT-4 win-rate Pareto frontier calculated?

---

### Official Review · Reviewer_W8Jh · 2025-10-31

**Soundness:** 3
**Presentation:** 3
**Contribution:** 3
**Rating:** 6
**Confidence:** 3

**Summary:**

The authors provide a novel method of combining multiple rewards to ensure the combination remains a BT model

**Strengths:**

The method is innovative and indeed reduces MORLHF into a BT preference model.

**Weaknesses:**

The authors should discuss the contrastive sample genreration in detail. Especially given that in a multiobjective setting finding a contrastive sample that can meaningfully contribute to all metrics is not trivial.

**Questions:**

See weakness.

The results seem to be on only two objectives: harmlessness and safety. It would be meaningful to see how the methodology adapts to more multi-objectives.

---

### Official Review · Reviewer_4CuH · 2025-11-03

**Soundness:** 3
**Presentation:** 2
**Contribution:** 3
**Rating:** 4
**Confidence:** 2

**Summary:**

The paper proposes a probability-based reward combination for multi-objective RLHF: convert each objective’s reward to its Bradley–Terry win probability, weight and aggregate in probability space, then apply logit to obtain a single reward for PPO/GRPO. This aims to reduce cross-preference interference and improve the Pareto frontier on safety/helpfulness across datasets and algorithms.

**Strengths:**

Clear motivation & simple drop-in design: identifies issues with linear value scalarization and offers a minimal change—probability-space fusion + logit—compatible with standard RLHF loops.

Probability based Reward
Works across training algorithms (BT-PPO, BT-GRPO) with consistent gains across weight settings; suggests scalability to larger preference spaces.

**Weaknesses:**

Baselines/Ablations: Comparisons are too weak. Add calibrated baselines (per-objective z-score/temperature or isotonic), logit-space/probability fusions (sum of logits; geometric vs. arithmetic mean), and multi-objective scalarizations (Chebyshev, ε-constraint). Clarify if gains come from scale calibration vs. probability-space design.

Generalization: The paper just show the result of two objectives. Considering add a third (e.g., honesty/factuality) and cross-domain datasets; report a 3D Pareto frontier or 2D projections.

Writing/Figures: Tighten captions and grammar, standardize terminology/capitalization, and improve figure readability (axes, legends). Ensure Figure 1 is unambiguous.

**Questions:**

Calibration vs. probability-space: If each reward is temperature-calibrated (or z-scored) first, do linear and BT methods close the gap? Please add those baselines.

Fusion variants: What about logit-space summation (∑wᵢ·logit pᵢ) or geometric vs. arithmetic means of probabilities—do results hold?

Theory: Can you formalize the “reduced cross-preference interference” claim with bounds (e.g., bias from non-commutativity) beyond qualitative Eq. 9 discussion?

---

### Meta-Review · Area_Chair_awCq · 2026-01-11

**Summary:**

The paper studies multi-objective alignment in RLHF and proposes a probability-based reward aggregation method based on Bradley-Terry (BT) model. The method combines multiple reward objectives using preference probabilities, rather than fixed linear weights at the reward-score level.

Strengths

- Introduces a new perspective on multi-objective alignment based on the BT preference model.

- The method is clearly motivated and appears simple to use in practice.

Weaknesses

- Inconsistency with the target BT formulation.

- Limited to two objectives.

- Experiments are restricted to medium-scale models and safety-alignment datasets. Missing comparisons to relevant baselines.

- Missing explanation of the contrastive sample generation process.

- Presentation quality is weak, with many equations not well-explained.

The authors did not provide a rebuttal. The raised concerns need to be addressed before acceptance.

**Reviewer Concerns:**

See weaknesses in Summary

**Reviewer Scores:**

Will not change.

---

### Decision · Program_Chairs · 2026-01-26

Reject